# Assessing Hydrological Simulations with Machine Learning and Statistical Models

**Evangelos Rozos** 

Institute for Environmental Research & Sustainable Development, National Observatory of Athens, 15236 Athens, Greece; erozos@noa.gr; Tel.: +30-210-810-9125

**Abstract:** Machine learning has been used in hydrological applications for decades, and recently, it was proven to be more efficient than sophisticated physically based modelling techniques. In addition, it has been used in hybrid frameworks that combine hydrological and machine learning models. The concept behind the latter is the use of machine learning as a filter that advances the performance of the hydrological model. In this study, we employed such a hybrid approach but with a different perspective and objective. Machine learning was used as a tool for analyzing the error of hydrological models in an effort to understand the source and the attributes of systematic modelling errors. Three hydrological models were applied to three different case studies. The results of these models were analyzed with a recurrent neural network and with the k-nearest neighbours algorithm. Most of the systematic errors were detected, but certain types of errors, including conditional systematic errors, passed unnoticed, leading to an overestimation of the confidence of some erroneously simulated values. This is an issue that needs to be considered when using machine learning as a filter in hybrid networks. The effect of conditional systematic errors can be reduced by naively combining the simulations (mean values) of two or more hydrological models. This simple technique reduces the magnitude of conditional systematic errors and makes them more discoverable to machine learning models.

**Keywords:** hydrological modelling; error analysis; systematic errors; machine learning; statistical models; recurrent neural networks; LSTM; Bluecat

## 1. Introduction

Currently, machine learning (ML) is taking its place among the standard options for simulating various hydrological processes [1,2]. The first use of an ML approach appeared almost 30 years ago [3] in a study that applied a recurrent neural network (RNN) [4] so that antecedent flow ordinates served to distinguish between the rising limb and the recession "and provide additional information about the input pattern". The complexity of employed ML models has increased significantly since then, with some recent applications employing complex networks with the number of weights and biases reaching up to 266,497 [5]. These sophisticated ML models have been proven to be superior to traditional hydrological models in studies that have performed comparisons over numerous water basins [6]. However, this higher performance comes with a cost of a very high computational burden (10 h to train a long short-term memory ensemble (LSTM) on a machine with 188 GB of RAM and a single NVIDIA V100 GPU [6]).

To avoid this disadvantage, some researchers have employed simpler ML models combined with hydrological models to achieve better performance. For example, Aparicio et al. [7] used an extreme learning machine [8] to estimate the instantaneous peak flow, for which the input was the maximum mean daily flow, simulated by the Soil and Water Assessment Tool (SWAT). Noymanee et al. [9] employed a multi-layer network with one hidden layer of 100 nodes to improve the flood forecasting efficiency of Mike11 simulations. Althoff et al. [10] used the output from the first components (soil moisture) of the modèle

du Génie Rural à 4 paramètres Journalier (GR4J) hydrological model [11] as inputs to cubist regression to achieve both superior performance and model explainability (the soil moisture controlled the water percolation, which was the main predictor of the streamflow).

Previous studies show that the use of ML for boosting the performance of hydrological models is promising. However, despite being much more lightweight than approaches based exclusively on ML models, they are still relatively complicated and require tailored models. The reason for the increased performance in hybrid models is because the ML module functions as a filter that catches and corrects systematic errors of the hydrological model. The question is, if there are systematic errors in the simulations of the hydrological model, why not try to fix them instead of "absorbing" them with another model? Even if fixing these errors is not feasible due to hydrological model limitations, being in a position to identify them, i.e., understand their attributes, would be valuable knowledge regarding modelling uncertainty.

Rozos et al. [12] employed an approach based on machine learning to detect systematic errors. The core concept of this approach is the best model achievable with the available data. It is symbolized as $\Phi(x)$ and defined as:

$$\Phi(x) = y_{\mathrm{m}}(z; \theta) + f(x; \theta) \tag{1}$$

$$\Phi(x) = y_{\mathrm{o}} - \epsilon \tag{2}$$

where $z$ is the inputs (i.e., the stresses) of the hydrological model $y_{\mathrm{m}}$, $x$ is the superset including the hydrological model inputs and outputs, $\theta$ is the vector with the parameters of the hydrological model, $f$ is the function that gives the systematic errors of the hydrological model, $y_{\mathrm{o}}$ are the observations and $\epsilon$ are the random errors.

Equation (2) can be used to obtain an approximation of the best model achievable with the available data by fitting an ML model to the observed data. In this case, the difference with the application of ML models in place of hydrological models is that the explanatory variables $x$ include not only the stresses, i.e., the hydrological model inputs, but also the hydrological model outputs. By substituting this approximation into Equation (1), an estimation of the systematic errors of the hydrological model (i.e., the function $f$) can be obtained. Another difference of this approach, compared with hybrid models where the ML serves as a filter that "absorbs" the systematic errors, is that the ML network is minimal with a fixed topology, which can be trained with minimal CPU requirements and time.

Another option to estimate systematic modelling errors is to employ statistical methods. Koutsoyiannis and Montanari [13] have suggested a method to estimate the local uncertainty of deterministic simulations based on a simple concept. They call this method Bluecat. For each assessed simulated value of the test period, a sampling of the most related observations of the calibration period is performed. This sampling includes observations that correspond to the simulated values of the calibration period that are closer to the assessed value of the test period. The empirical distribution of the sampling set provides valuable information regarding the uncertainty of the hydrological model attributed to all types of errors. Thus, if, for example, a hydrological model consistently overestimates the flow, then the median will be systematically lower than the simulated values. The median can be used as a statistical prediction based on the hydrological simulation, and this is actually another method to approximate the best model achievable with the available data. However, this method was not used in this study because its accuracy is compromised in high flows where the number of observations are limited, which introduces sampling bias (this issue has been addressed by a recent work of Koutsoyiannis and Montanari [14]).

In this study, these two approaches to analyze the systematic errors of hydrological models were applied to three different case studies with daily (the first) and hourly time steps (the other two). This allowed us to derive some general conclusions regarding the capacities of these two methods. Some interesting findings regarding the conditional systematic errors, and guidelines on how to cope with them are also provided.

## 2. Materials and Methods

### 2.1. Hydrological Models

Three hydrological models were employed in this study: HYdrologic MODel (HYMOD) [15], GR4J/H [11] and LRHM [16]. All three models are conceptual models. See the Data Availability Statement for details on how to obtain these models.

HYMOD employs 5 parameters (Figure 1): the maximum storage capacity in the catchment Cmax, the ratio of the diversion to quick- and slow-release reservoirs, the factor Bex,p which defines the variability in space of the surface soil moisture capacity, the outflow coefficient Kq of the quick water storage tank, and the outflow coefficient Ks of the slow tank. HYMOD simulations were performed with a model version implemented in R programming language. The parameters were optimized with the DEoptim R package, which implements the differential evolution algorithm for global optimization of a real-valued function of a real-valued parameter vector [17].

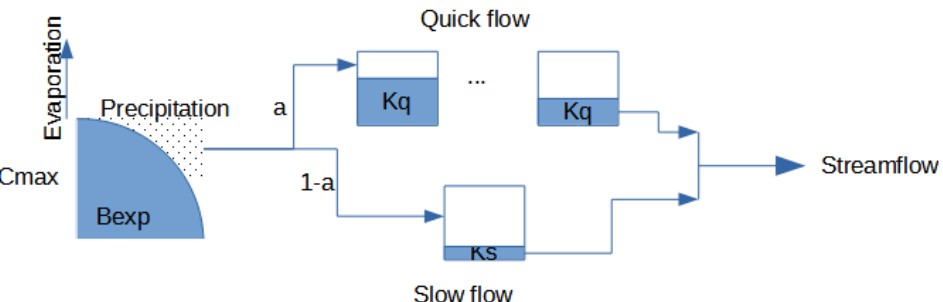

**Figure 1.** Schematic representation of the HYMOD model.

LRHM is based on two simple model building blocks (direct runoff and soil moisture model, Figure 2) that are linearly combined to simulate the observed runoff (an idea related to the genetic programming model building [18]). If all building blocks are activated, 16 parameters are employed, four for each of the two soil moisture components (k1, c1, k2, c2), three scaling factors ($\alpha$, $\beta$, $\gamma$) and five coefficients (a0, a1, a2, a3, a4) of the linear regression. LRHM was implemented in MATLAB (compatible with GNU Octave). The parameters were optimized with Genetic Algorithm [19].

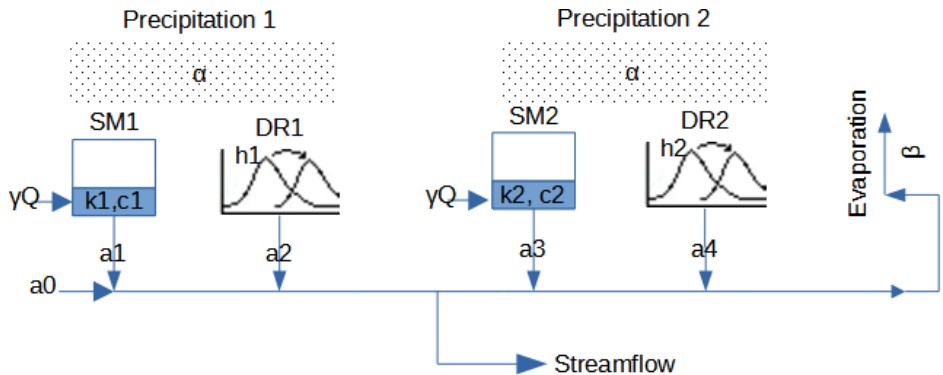

**Figure 2.** Schematic representation of the LRHM model.

GR4J employs only four parameters (Figure 3): ×1 is the maximum capacity of the soil moisture accounting store, ×2 is the water exchange coefficient of the groundwater exchange term, ×3 is the reference capacity and ×4 is the time parameter of the unit hydrographs. GR4H (hourly time step) uses the same equations as GR4J (daily time step). The different time step is reflected by a change in the parameter values, which depend on time [20]. The GR4J simulations were run employing the airGR package in the R programming language [21]. The parameters were optimized with the Michel algorithm [22].

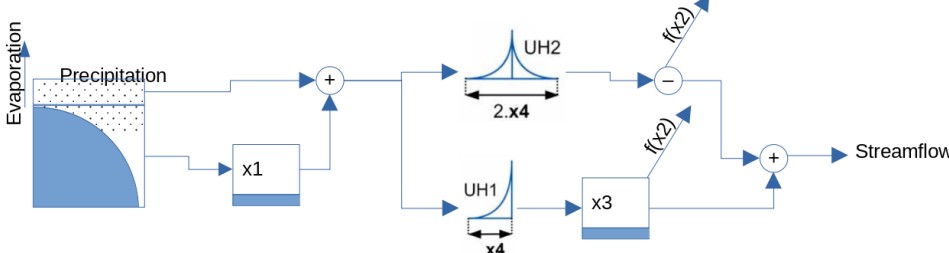

**Figure 3.** Schematic representation of the GR4J model.

The Nash–Sutcliffe efficiency index [23] was used as an objective function for the calibration of all three hydrological models.

### 2.2. Recurrent Neural Network

The network topology is displayed in Figure 4. It is a recurrent neural network (RNN) employing long short-term memory (LSTM) [24]. The three input nodes correspond to the time series of rainfall, evaporation and simulated values from the hydrological model. The network has 6 nodes in the hidden layer (twice the number of inputs). A single node was used as the output, which is the approximation of the best model achievable with the available data (see Equations (1) and (2)). It is noted that the topology of Figure 4 is slightly different from that used in [12], for which an equal number of nodes in the input and hidden layers was used. This modification was made due to slightly better performance observed with the setup of 6 nodes. The activation function between input and hidden layers is LReLU, and between hidden and output layers ReLU [25].

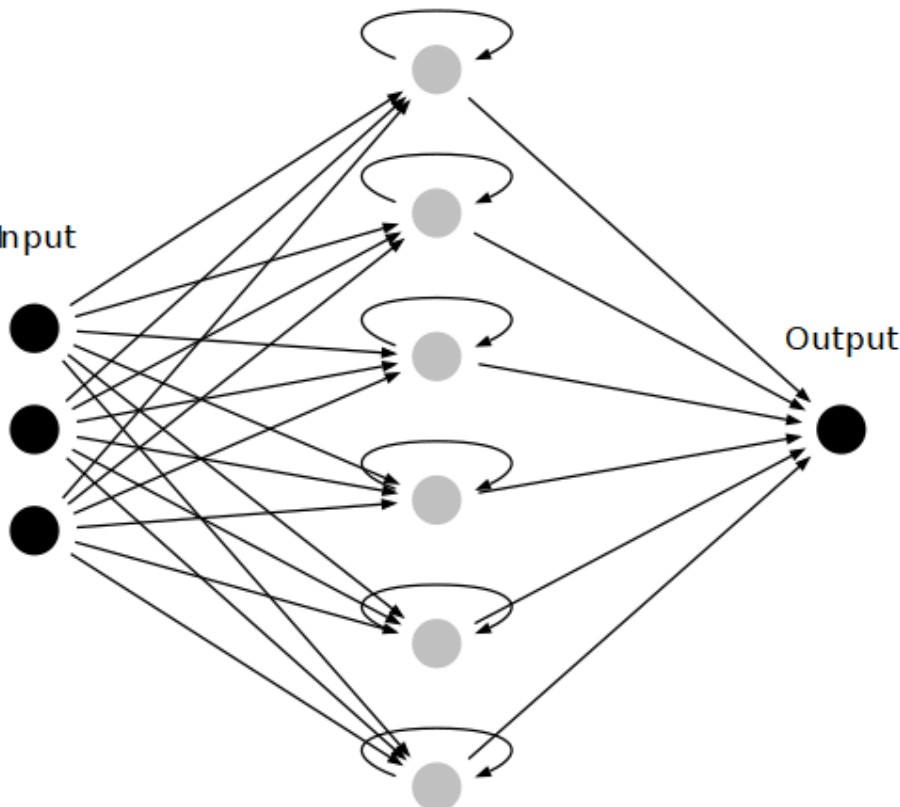

**Figure 4.** The topology of the machine learning network.

The mean squared error was used as the loss function. The ML network was trained with the gradient descent algorithm ADADELTA [26], without employing mini-batch. Z-score normalization [27] was used to scale the input data (minmax was also tested, but did not offer any improvement). To avoid overfitting [28], early stopping regularization was employed, which has been found to be efficient in applications of LSTM for predicting stream flows [29].

Cortexsys [30] (a deep learning toolbox for MATLAB and GNU Octave) was employed for the implementation of the network. The machine learning network was also implemented with Keras [31] in a Google Colaboratory workbook to make it available without requiring the installation of any software (see Data Availability Statement).

*2.3. KNN-Bluecat*

The statistical method used to assess the hydrological models was based on the work of Koutsoyiannis and Montanari [13]. They call this method Bluecat. In their study, they used the conditional distribution of a true (observed) value given a simulated value as the measure of the uncertainty of a hydrological model . This conditional distribution can be estimated with a conditional probability according to the following formula:

$$F_{\underline{q}|\underline{Q}}(q|Q) \approx P\{\underline{q} \leq q \mid Q - \Delta Q_1 \leq \underline{Q} \leq Q + \Delta Q_2\} \tag{3}$$

where $q$ and $Q$ are the random variables that correspond to the observed and simulated values $q$ and $\bar{Q}$, respectively, and $\Delta Q_1$ and $\Delta Q_2$ define a neighbourhood of Q such that the intervals above and below Q contain appropriate numbers of simulation values, such as $2m + 1$ (for unbiased sampling, the closest plus an equal number of $m$ values above and below $Q$ should be selected).

Rozos et al. [32] indicated that Equation (3) is equivalent to applying the empirical distribution function on the k-nearest neighbours obtained with the KNN method where $k = 2m + 1$ [33] (see Figure 5). We call this approach KNN-Bluecat and used it for convenience, instead of the original Bluecat, in this study (see Data Availability Statement).

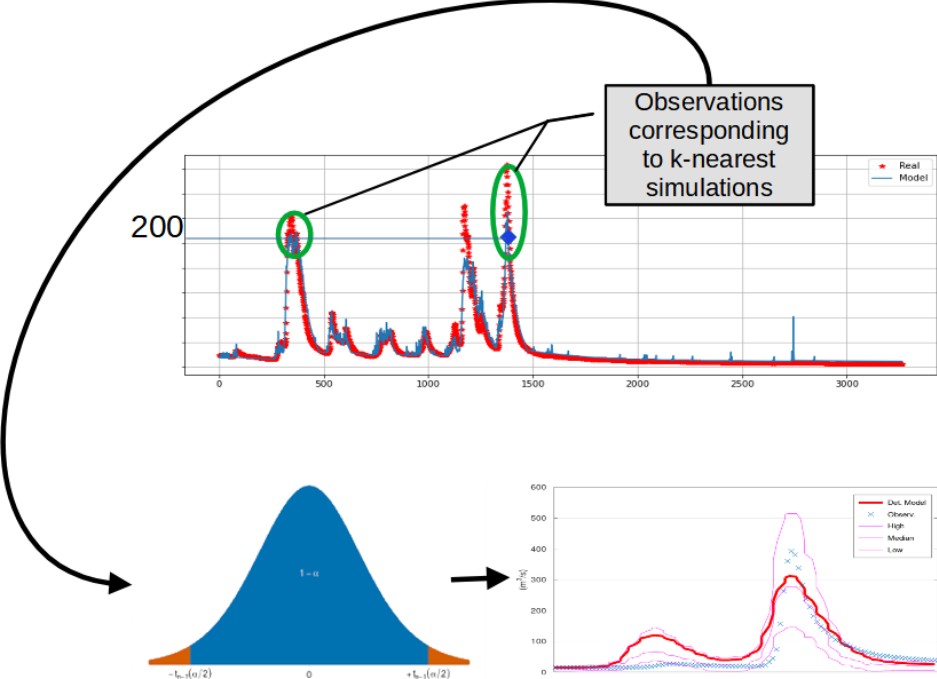

**Figure 5.** Schematic representation of KNN-Bluecat. The red dots are the observed values during the calibration period. The assessed simulated value during the test period is 200. The empirical distribution of the k observations that correspond to the k nearest to the assessed simulated value (200) gives information regarding the uncertainty of the model.

*2.4. Performance Indicators*

Two metrics were employed to evaluate hydrological model simulations: mean squared error (MSE), Equation (4), and percentage of bias (PBIAS), Equation (5).

$$\text{MSE} = 100\frac{\sum_{i=1}^{n}(P_i - O_i)^2}{n}, \quad 0 \le \text{MSE} < +\infty \tag{4}$$

$$\text{PBIAS} = 100\frac{\sum_{i=1}^{n}(P_i - O_i)}{\sum_{i=1}^{n}O_i}, \quad -\infty < \text{PBIAS} < +\infty \tag{5}$$

where $n$ is the length of the time series, $P_i$ is the prediction at time step $i$ and $O_i$ is the observation at time step $i$.

In general, these two metrics are common in hydrological applications [10,34] and complement each other well. For example, MSE places more emphasis on the errors of larger values, but tends to overlook overestimations of low values. On the other hand, PBIAS is more sensitive to consistent over/underestimation of the observed values.

*2.5. Case Studies*

The previously mentioned methods (RNN with LSTM and KNN-Bluecat) were tested in the following locations:

- Arno River at Subbiano, Tuscany, Italy. The catchment of Arno River is 752 km$^2$. The observed data include the mean areal daily rainfall, evapotranspiration, and discharge at the basin exit. The period of the available data starts on 2 January 1992 and ends on 1 January 2014 (8037 time steps). The annual rainfall was 1213 mm/year.
- Sieve River at Fornacina, Tuscany, Italy. The catchment area of Sieve River is 846 km$^2$. The observed data include the mean areal hourly rainfall, evapotranspiration and discharge at the basin exit. The period of the available data starts on 3 June 1992 and ends on 2 January 1997 (36,554 time steps, with a gap in the data from 1 January 1995 to 2 June 1995). The flow regime of Sieve River is intermittent. The annual rainfall was 1190 mm/year.
- Bakas River, tributary of Nedon River, Messenia, Greece. The catchment area of Bakas River is 90 km$^2$. The average annual precipitation depth is 1000 mm. The simulation time step was hourly, with the observations extending from 1 September 2011 01:00 to 1 May 2014 00:00 (23,353 time steps). The annual rainfall was 1393 mm/year.

For all the case studies, 70% of the available data were used for the calibration of the hydrological model and the remaining data were used for model validation (the split of the training/test sets of the three case studies was: Arno 5626/2410, Sieve 27,415/9138 and Bakas 15,732/6742). The training and test periods of the ML models coincided with the previous two periods. More information regarding these locations (maps and description of geomorphological characteristics) can be found in [13] for the first two case studies and in [16] for the last case study .

## 3. Results

*3.1. Case Study—Arno*

Table 1 gives the performance indicators of the simulations with the three hydrological models and LSTM predictions. Figure 6 displays the simulations and predictions for the time steps 1900 to 2120 of the test period.

**Table 1.** Performance indicators of Arno River case study.

|  |  | LSTM | Hydrological Model |
|---|---|---|---|
| HYMOD | MSE of training period | 94.58 | 233.77 |
|  | PBIAS of training period | −3.05% | 33.70% |
|  | MSE of test period | 195.36 | 299.25 |
|  | PBIAS of test period | 7.72% | 44.19% |
| LRHM | MSE of training period | 128.80 | 143.59 |
|  | PBIAS of training period | 1.85% | −2.39% |
|  | MSE of test period | 197.69 | 198.25 |
|  | PBIAS of test period | 14.30% | 10.03% |
| GR4J | MSE of training period | 60.25 | 105.12 |
|  | PBIAS of training period | −0.22% | −3.99% |
|  | MSE of test period | 164.99 | 165.56 |
|  | PBIAS of test period | 11.07% | 7.75% |

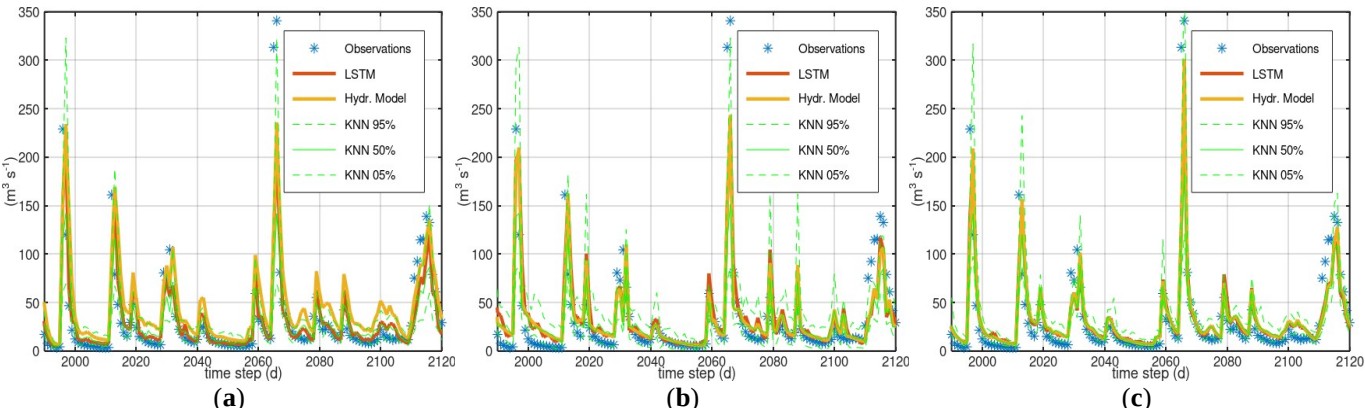

(a)　　　　　　　　　　　　　　　　(b)　　　　　　　　　　　　　　　　(c)

**Figure 6.** Simulation of Arno River, for time steps 1990–2120 of the test period: (**a**) HYMOD (note how LSTM avoids the overestimation of the low flows); (**b**) LRHM; and (**c**) GR4J.

HYMOD—This model overestimated the average flow, which resulted in a significant bias and MSE. The LSTM predictions significantly reduced this bias and had lower MSE in both training and test periods. The width of the CI seemed to capture the model error and had a plausible shape. The values of HYMOD simulations were occasionally on the upper limit of the CI.

LRHM—The LSTM predictions only slightly outperformed the LRHM simulations of the training period. Regarding the test period, the MSE of the LSTM predictions was similar to that of the LRHM simulations, whereas they had a greater bias. The shape of the CI was plausible and its width, for low flows, was the largest among the three models. The LRHM simulations and LSTM predictions were close, especially for midrange to low flow values.

GR4J—This hydrological model had the best performance among all three models. The LSTM predictions had much lower MSE in the training period than the hydrological model, whereas they had similar MSE and higher bias in the test period. The shape of the CI was plausible. The GR4J values and LSTM predictions coincided, indicating that LSTM did not detect any systematic errors.

### 3.2. Case Study—Sieve

Table 2 gives the performance indicators of the simulations with the three hydrological models and the LSTM predictions. Figure 7 displays the simulations and predictions for the time steps 900 to 2000 of the test period whereas Figure 8 for the time steps 8080 to 8180 of the test period.

**Table 2.** Performance indicators of Sieve River case study.

|  |  | LSTM | Hydrological Model |
|---|---|---|---|
| HYMOD | MSE of training period | 170.07 | 277.45 |
|  | PBIAS of training period | −3.90% | 30.99% |
|  | MSE of test period | 284.21 | 417.89 |
|  | PBIAS of test period | 7.50% | 42.12 % |
| LRHM | MSE of training period | 163.31 | 198.77 |
|  | PBIAS of training period | 2.42% | 3.64% |
|  | MSE of test period | 301.91 | 338.92 |
|  | PBIAS of test period | −3.89% | −0.07% |
| GR4H | MSE of training period | 89.15 | 121.25 |
|  | PBIAS of training period | −0.73% | 10.55% |
|  | MSE of test period | 129.60 | 147.55 |
|  | PBIAS of test period | 7.53% | 19.48% |

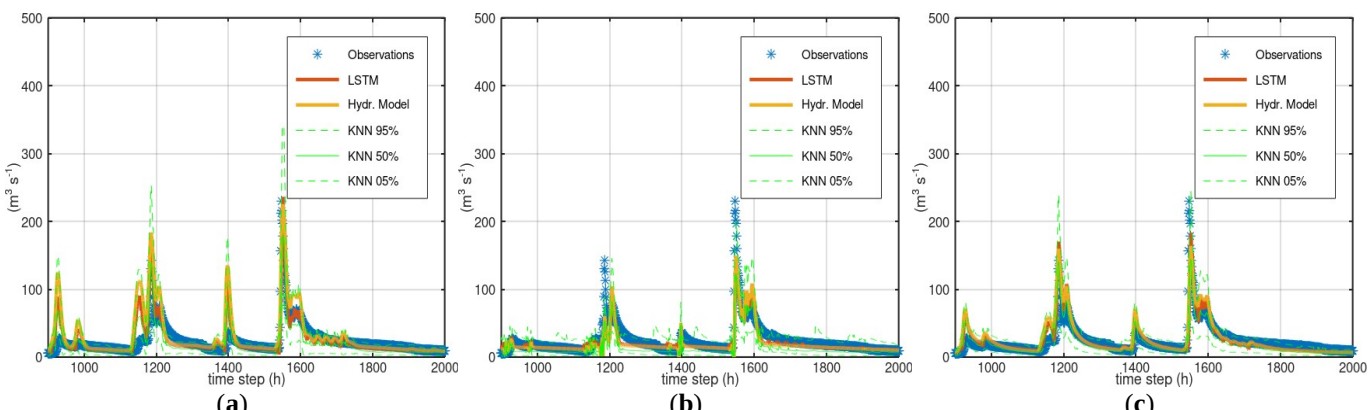

**Figure 7.** Simulation of Sieve River, for time steps 900–2000 of the test period: (**a**) HYMOD (note how LSTM moderates the erroneous peak flows); (**b**) LRHM (note that LSTM does not repair the occasional failure of the hydrological model at high flows); and (**c**) GR4H.

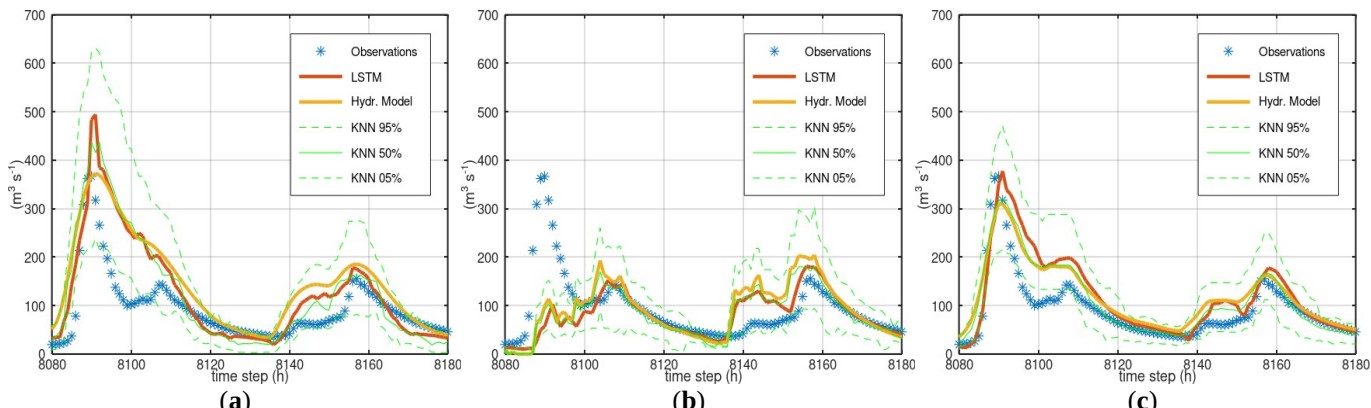

**Figure 8.** Simulation of Sieve River, for time steps 8080–8180 of the test period: (**a**) HYMOD; (**b**) LRHM (note the confidence interval between time steps 8080 and 8100); and (**c**) GR4H.

HYMOD—HYMOD tended to overestimate the flows and occasionally gave erroneous midrange flows when the corresponding observed flow was minimal (Figure 7). LSTM predictions corrected these errors to some extent and reduced the MSE and bias in both the training and test periods. However, they did not eliminate the erroneous peaks, which showed that LSTM did not successfully detect the magnitude of the systematic error. Though HYMOD correctly simulated the peak at time step 8090 of the validation period

(Figure 8), the CI was the widest of the whole validation period. Overall, the CI shape was plausible, with the interval getting wider at periods with higher flows, and vice versa. LSTM indicated a systematic overestimation over the whole range of flows. If this case study was characterized as extreme, with an assumed system capacity equal to 200 $m^3/s$, HYMOD successfully simulated this exceedance in five out of the six observed events and gave one falsely simulated exceedance.

LRHM—LRHM occasionally failed to reproduce high flows and LSTM predictions did not improve this underestimation. LSTM predictions were slightly better during both the training and the test periods, but had a slightly higher bias during the test period. LRHM failed to reproduce the peak at time step 8090 (Figure 8). Despite this, the width of the CI at this time step was quite narrow, and the peak of the observed flow was far outside of this interval. During the test period, LRHM successfully simulated the exceedance in only two out of the six events of high flow and gave one falsely simulated exceedance of 200 $m^3/s$.

GR4H—GR4H had the lowest MSE among all models, but it had significant bias. The LSTM predictions were better during both the training and test periods. GR4H successfully simulated the exceedance in four out of the six events of high flow and gave no falsely simulated exceedance of 200 $m^3/s$.

### 3.3. Case Study—Bakas

Table 3 gives the metrics of the simulations with the three hydrological models and LSTM predictions.

**Table 3.** Performance indicators of the case study of Bakas River.

|  |  | LSTM | Hydr. Model |
|---|---|---|---|
| HYMOD | MSE of training period | 2.0923 | 2.5895 |
|  | PBIAS of training period | −6.92% | 2.52% |
|  | MSE of test period | 2.9315 | 2.8848 |
|  | PBIAS of test period | 5.79% | 20.25% |
| LRHM | MSE of training period | 1.3191 | 1.9080 |
|  | PBIAS of training period | −0.15% | 0.39% |
|  | MSE of test period | 2.9227 | 3.2485 |
|  | PBIAS MSE of test period | −11.13% | −14.14% |
| GR4H | MSE of training period | 1.5243 | 2.4389 |
|  | PBIAS of training period | 6.49% | 4.91% |
|  | MSE of test period | 3.1121 | 3.2996 |
|  | PBIAS MSE of test period | −3.09% | −16.69% |

HYMOD—The LSTM predictions had lower a MSE but higher bias than the hydrological model during the training period, and a similar MSE and much smaller bias during the test period. The shape of the CI presented some peculiar features. For example, the lower limit presented a sudden increase and the upper limit presented a gradual decrease a little after time step 1800 (Figure 9). At the same time, the simulated values presented a linear decrease. This shape gave non-realistic readings regarding model uncertainty. For example, for a simulated value equal to 1.95 $m^3/s$ (time step 1800), the lower and upper limits of the 90% CI were 0.3 and 6.86 $m^3/s$, respectively, whereas for a simulated value equal 1.72 $m^3/s$ (time step 1950), the lower and upper limits were 1.54 and 3.56 $m^3/s$, respectively. Note that the corresponding empirical CDF values of the observations and simulated time series did not indicate any abrupt changes in this region (the empirical CDF values of the observed and simulated time series for the value 1.95 $m^3/s$ were 0.62 and 0.61, respectively, and for the value 1.72 $m^3/s$, 0.61 and 0.58, respectively).

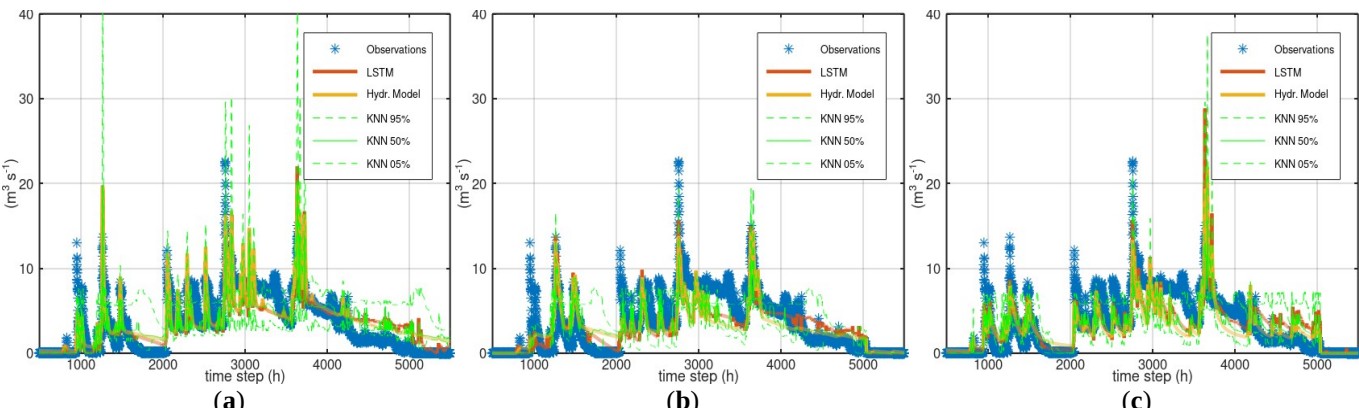

**Figure 9.** Simulation of Bakas River, for time steps 500–5500 of test period: (**a**) HYMOD; (**b**) LRHM; and (**c**) GR4H (note the "preference" of the upper limit of the CI for a specific value (around 6 m³/s)).

LRHM—The LSTM predictions were more accurate than the hydrological model simulations during both training and test periods. The LSTM occasionally provided an accuracy improvement (e.g., the period between 1500 and 2000 h of the validation period, see Figure 9), but in some cases, its predictions were worse than the model simulation values (e.g., low flows between 2000 and 2500). It was hard to distinguish the pattern of the systematic error. LSTM did not detect the failure of the hydrological model to capture the high flows around time step 1000 (Figure 10), and the CI did not give a correct estimation of the error at this location. As in the analysis of the HYMOD simulation, the shape of the CI exhibited the same interesting features during time steps 1500 to 2000. Furthermore, a peculiar increase in the CI width took place during low flows between time steps 2000 and 2500, when the hydrological model actually simulated the flows rather well. This increase of the width during low flows is counter-intuitive, as the CI is expected to get narrower at lower flows.

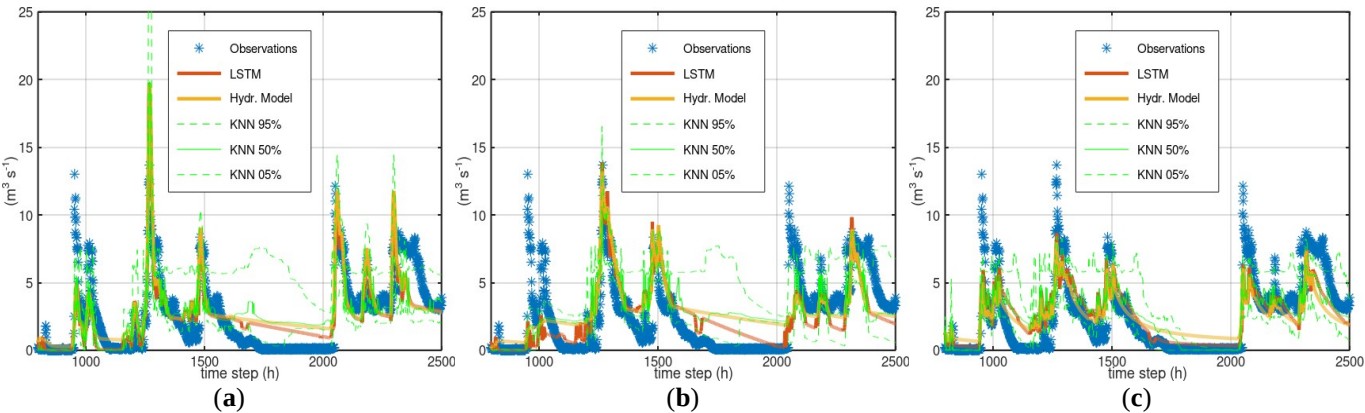

**Figure 10.** Simulation of Bakas River, for time steps 800–2500 of test period: (**a**) HYMOD (note the CI width around time step 1800 and the upward step of the lower limit right after); (**b**) LRHM (note the increase in CI width between time steps 2000–2500 during low flows); and (**c**) GR4H (note the "preference" of the upper limit for a specific value (around 6 m³/s)).

GR4H—The LSTM predictions were more accurate than the hydrological model simulations for both the training and test periods. The CI did not present the non-realistic features of the previous two models. However, the upper limit tended to fluctuate frequently around a specific value (6 m³/s).

## 4. Discussion

The application of the LSTM and KNN-Bluecat in three case studies employing three hydrological models (HYMOD, LRHM and GR4H/J) allowed us to derive conclusions

regarding their weaknesses and strengths in assessing hydrological model simulations. The purpose of employing LSTM was to investigate the existence of systematic errors, whereas the purpose of employing KNN-Bluecat was to estimate the model uncertainty.

LSTM was proven to be very successful in detecting whether there was a bias that could be reduced without deteriorating the overall accuracy (see case studies of Arno River and Sieve River). The requirement to not compromise overall accuracy was essential, as zero bias can easily be achieved with a constant value equal to the mean of the observations. Moreover, the LSTM predictions moderated the erroneous simulations of midrange flows whenever the observed flow was minimal. From these findings, it can be inferred that LSTM can detect the systematic errors related to overestimation of flows, though it cannot accurately estimate their magnitude.

LRHM failed to reproduce some events in the case study of the Sieve River (see Figures 7b and 8b), for which LSTM did not indicate that the LRHM value was erroneous. It is not clear why this occurred. It may have been because there was no sufficient signal strength from similar incidences during the training period to allow LSTM to learn from it. That is, during the training period, there were only three events with a flow greater than 100 m$^3$/s that LRHM fails to simulate. These three events corresponded to around 60 observations of the training data out of a total of 27,415 observations. However, the sparsity of high flows in the data is typical in hydrological applications. Therefore, the value of LSTM in detecting the errors of a single hydrological model in high flows is disputable.

Regarding KNN-Bluecat, the CI shape and width were plausible in the first two case studies (the upper and lower limits followed the fluctuation of the simulated values, and the width followed the magnitude of the simulated values), but presented some peculiar features (increased CI width at low flows, discontinuities of the CI lower limit and "preference" of the CI upper limit for specific values) in the Bakas case study for all hydrological models employed. The reason was most likely the unrealistic steady flow that appeared between time steps 10,000–11,700 of the training period. Figure 11 displays the LRHM simulation of the two events of the training period. All hydrological models (LRHM is displayed in Figure 11) presented a recession curve instead of a steady flow during this period (i.e., this steadiness was not justified by the stresses). The peculiar shapes of the CI reflect the inconsistencies between the responses during time steps 1–4500 (Figure 11a) and the responses during time steps 10,000–11,700 (Figure 11b).

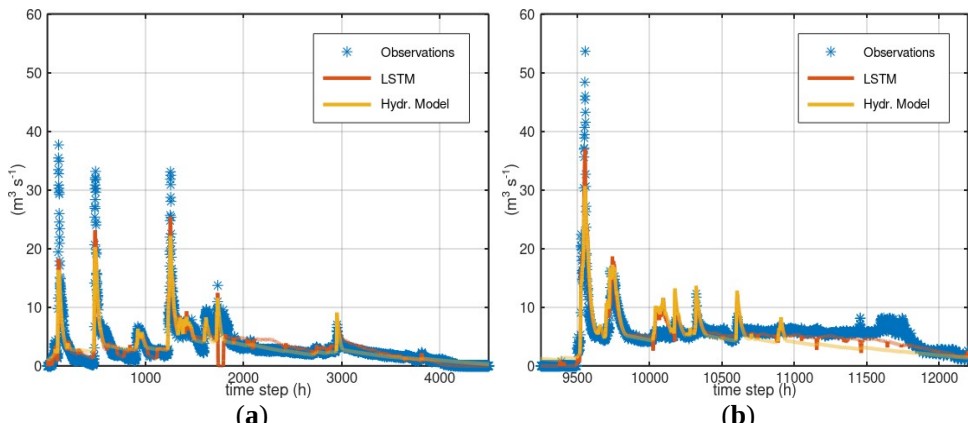

**Figure 11.** Simulation of Bakas River training period for (**a**) time steps 1–4500 and (**b**) time steps 9250–12,200 (note the observed steady flow between time steps 10,000 and 11,500).

Regarding the CI of LRHM in the Sieve River case study, the observed values of the training period that correspond to the simulated values of the training period similar to that of time step 8090 of the test period (100 m$^3$/s instead of the observed 380 m$^3$/s, see Figure 8b), were ranging from 50 to 150 m$^3$/s. This means that a similar error is not manifested in the training period. Therefore, though this significant discrepancy must be systematic (as the other two hydrological models have a much smaller error at this time

step), it may be some kind of condition that triggered it, which was not present during the training period.

Finally, it should be noted that the improvement achieved by LSTM in some cases was inferior to the performance of a plain hydrological model. For example, in the case study of the Sieve River, the MSE of LSTM was 284.21 when applied to HYMOD, whereas the MSE of GR4H alone was 147.55. This means that LSTM could not deliver the best achievable performance with the available data. This was also noted in [12], attributing it to the difficulty in obtaining an approximator that would yield an uncorrelated, homoscedastic and zero-inflated error.

To cope with the previous issues, we examined what benefits could be obtained by combining two or more hydrological models. The idea behind this concept was that by combining two or more hydrological models, the conditional errors (errors that happen under certain conditions, and hence, may not be present in the training period) may be reduced, which would give a chance to LSTM and KNN-Bluecat to capture them. The model simulations were combined by taking the mean values (more sophisticated methods, such as using both models as the inputs of the ML network were not proved advantageous). It is noted that approaches based on multiple models (model ensembles) combined with Bayesian averaging have been successfully applied by other researchers [35].

Figure 12 displays, for the three case studies, the MSE of the HYMOD simulations (1st bar of each panel), the MSE of LSTM using HYMOD simulations as the input (2nd bar of each panel), the MSE of LSTM using the mean of the HYMOD and LRHM simulations as input (3rd bar of each panel), and the MSE of LSTM using the simulations of the best hydrological model (4th bar of each panel). GR4J and GR4H achieved the best performance in the case studies of Arno River and Sieve River, whereas LRHM achieved the best (though marginally better than the other two) performance for Bakas River.

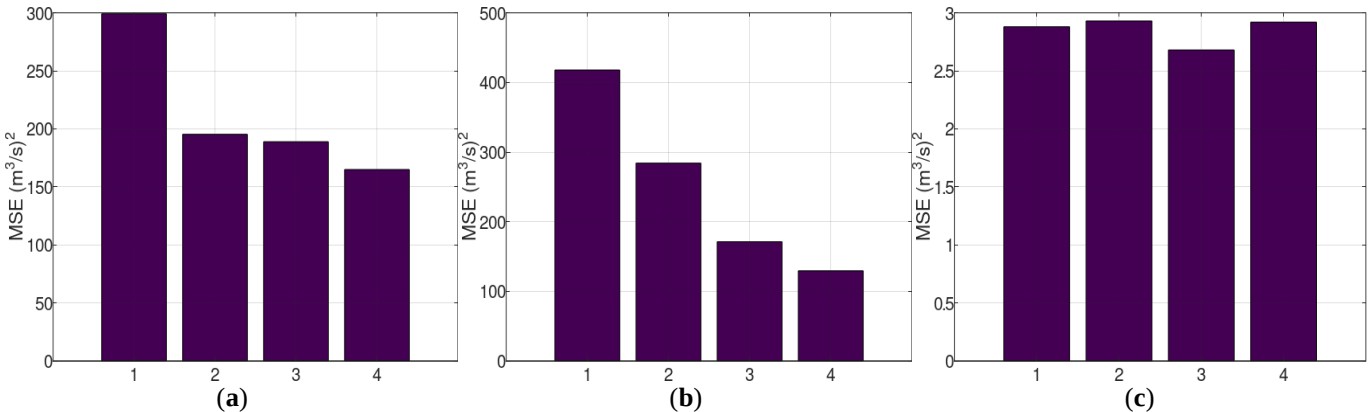

**Figure 12.** (**a**) Case studies of the Arno River, (**b**) Sieve River and (**c**) Bakas River. In all three panels: MSE of the HYMOD simulations (1st bar), LSTM with HYMOD simulations (2nd bar), LSTM with HYMOD+LRHM simulations (3rd bar) and LSTM with simulations of the best hydrological model (4th bar).

Supposing the 4th bars in the panels of Figure 12 give the best performance that can be achieved by a hydrological model, which we do not know, what could be inferred from the evolution of the MSE given by the first three bars? In the Arno River case study (Figure 12a), the MSE given by the 2nd bar is similar to that of the 3rd bar, and both are much lower than that of the HYMOD. This is an indication that the best achievable performance has been reached. Indeed, these bars are similar to the 4th bar, which we have assumed gives the best achievable performance. In the Sieve River case study (Figure 12b), the height of the bars from the 1st to the 3rd bar keeps decreasing. Most likely, the best achievable performance has not been reached. However, the performance that corresponds to the 3rd bar (i.e., LSTM predictions when using the mean of HYMOD and LRHM) is much better than that of the 1st (HYMOD simulations) and much closer to the 4th (LSTM predictions

when using the best hydrological model in this case study, which was GR4H). In the Bakas River case study (Figure12 c), the first three bars all have the same height, which is a strong indication that the best achievable performance has been reached, and this is verified by the 4th bar.

It is worth noting that in the case study of the Sieve River, the LSTM performance with inputs the simulations of HYMOD and LRHM (not their mean, separately) was 284.21 and 301.91 $(m^3/s)^2$, respectively, making them marginally better than the performance of LRHM alone. On the other hand, the performance of LSTM with inputs for the combined simulations of HYMOD and LRHM was much better, i.e., 171.29 $(m^3/s)^2$. This indicates that the technique suggested by some researchers to use an RNN as a filter to absorb systematic errors of a single model, despite improving performance, may still fall short from what can be achieved with the available information.

The information that can be obtained after assessing hydrological models with KNN-Bluecat and LSTM network is summarized in Table 4.

**Table 4.** Assessing hydrological models (test period) with KNN-Bluecat and LSTM.

| Tool | Diagnostic | Interpretation |
|---|---|---|
| KNN-Bluecat | Unusual CI shape and/or width | Inconsistencies in data |
| KNN-Bluecat | Narrow CI width despite large error | Conditional systematic errors |
| KNN-Bluecat | Model simulation far from median | Model bias |
| LSTM | Similar performance of LSTM on multiple hydrological models and on plain model | Best possible performance achieved |
| LSTM | Similar performance of LSTM on multiple hydrological models and on single model, but better than that of plain model | Best achievable performance detected, hydrological model falls short |
| LSTM | Performance of LSTM on multiple hydrological models better than that on single model and plain hydrological model | Best achievable performance not detected, hydrological model falls short |

It should be noted that these assessments of the performance of the models are based on the use of a single performance metric (in this case, the MSE). However, a model can present a much lower MSE than another model despite exhibiting obvious errors (for example, systematic bias in low flows). For this reason, the results of the suggested methodology should be interpreted along with the overall behaviours of the models.

## 5. Conclusions

In this study, the errors of a hydrological model were analyzed employing machine learning and statistical techniques. The objective was to identify how far the performance of the assessed model was from the best achievable performance. Three hydrological models (HYMOD, LRHM and GR4J/H) were applied to three different case studies. The findings can be summarized by the following:

- Statistical approaches that estimate the model uncertainty based on observations (e.g., Bluecat or KNN used in this study) can provide, besides an uncertainty analysis, an evaluation of the consistency of the available data, i.e., the plausibility of the observed responses based on the observed stresses. Nevertheless, statistical approaches can underestimate the uncertainty if the assessed hydrological model exhibits conditional systematic errors.
- A simple recurrent neural network such as LSTM can be applied to the model results to detect systematic errors. In these case studies, it was efficient in detecting the systematic overestimations of hydrological models, but less reliable in detecting the failures of hydrological models at high flows. Conditional systematic errors appear to also escape the notice of machine learning approaches.
- A naive combination (mean of the simulated values) of the results of two hydrological models that simulate the same water basin offers the advantage of reducing the effect

of the systematic error of the models (especially the conditional systematic error). A recurrent neural network, such as LSTM, can be applied to a naive combination of the models' results to obtain a good approximation of the best achievable performance with the available data.

From these findings, it is apparent that employing multiple hydrological models is advantageous. Currently, there is a great variety of options including freely available models that require minimal time to set up and run. The naive combination (mean value) of the results of the models can reduce the systematic error, which allows a more reliable analysis of uncertainty and error. Combining multiple hydrological models with a recurrent neural network may be the best option for hybrid hydrological frameworks.

**Funding:** This research was funded by the Internal Grant/Award of the National Observatory of Athens "Low computational burden flood modelling in small to medium-sized water basins in Greece"—5080. The APC was funded by the Internal Grant/Award of the National Observatory of Athens "Low computational burden flood modelling in small to medium-sized water basins in Greece"—5080.

**Data Availability Statement:** Data of the case studies of the Sieve River and Arno River and the HYMOD model can be found at https://github.com/albertomontanari/hymodbluecat, accessed on 4 January 2023. GR4J/H can be found at https://webgr.inrae.fr/en/software/airgr/, accessed on 4 January 2023. LRHM can be found at http://hydronoa.gr, accessed on 4 January 2023. The KNN used to estimate model uncertainty and cast a statistical prediction (the median) can be found at http://hydronoa.gr, accessed on 4 January 2023. The ML model used to assess the performance of hydrological models can be found at http://hydronoa.gr, accessed on 4 January 2023.

**Conflicts of Interest:** The author declares no conflict of interest. The funders had no role in the design of the study; in the collection, analyses, or interpretation of data; in the writing of the manuscript; or in the decision to publish the results.

## Abbreviations

The following abbreviations are used in this manuscript:

| | |
|---|---|
| CI | Confidence interval |
| GR4J | Géenie Rural à 4 paramètres Journalier |
| HYMOD | Hydrological model |
| KNN | K-nearest neighbours |
| LSTM | Long short-term memory |
| LRHM | Linear regression hydrological modelling |
| ML | Machine learning |
| MSE | Mean squared error |
| PBIAS | Percentage of bias |
| RNN | Recurrent neural network |

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
