# Peer review of "Assessing Hydrological Simulations with Machine Learning and Statistical Models"

_hydrology, doi:10.3390/hydrology10020049_

Round 1
Reviewer 1 Report
Reviewer highly appreciates the authors for this work. The article is very good and adds to the new trends in hydrological simulation by using Machine Learning and Statistical Models. However, there are some minor comments, which the authors must make to improve the paper as follows:
1- In this study, three hydrological models were applied to three different case studies, please add the location map for these areas (Arno River at Subbiano, Tuscany, Italy; Sieve River at Fornacina, Tuscany, Italy and Bakas River, tributary of Nedon River, Messenia, Greece).
2- These models have been applied almost to the same climatic environment, whether in Italy or Greece, can these models be built and applied to a different environment, for example Africa?
Author Response
Please see attached pdf file.

Reviewer 2 Report
Dear Author,
I have had the pleasure to review your manuscript entitled "Assessing hydrological simulations with machine learning and statistical models", for publication in the MDPI Hydrology Journal.
Based on my review, my feelings regarding the manuscript are totally positive. I have found the paper very interesting, well written, moslty properly formatted, with a sound and solid contents. The topic is of interest, introduction is well crafted, the methodology is appropriate and supports well the findings. Also, the discussion is quite complete.
I have no major concern with the contents, but only minor suggestions that I have laid out in the form of comments within the manuscript. I am expecting point by point answers and actions taken for each of these comments.
Congratulations on such a good work. I recommend accepting the paper after minor revisions.

Author Response
Please see attached pdf file in response to Reviewer 1.
Reviewer 3 Report
This work uses ML as a diagnostic tool for the performance of various hydrological models. The work is novel and provides a perspective of how data-driven analysis can be used to assess and even improve conceptually hydrological models. Overall, the manuscript is well organised.
Few minor suggestions:
1. L82-86 Please give more details about the specific hydrological models used in the current work at the end of the introduction. The kNN-Bluecat and the RNN are presented but there is no information about the three hydrological models. I would expect a comparison between the three. Why were these models chosen in the first instance?
2. Section 2.2 – Please discuss the tuning of hyperparameters in this section.
3. L124-125 what was the metric used to assess the performance? Was it the MSE which was used for the rest of the analysis?
4. L176-177 I suggest that the author gives more details on training/test data for each location (time steps corresponding to test and training sets for each case).
Author Response

(The authors gave the same response as above.)
